# Tumor-Associated Macrophages—Implications for Molecular Oncology and Imaging

**DOI:** 10.3390/biomedicines9040374

**Published:** 2021-04-02

**Authors:** Melanie A. Kimm, Christopher Klenk, Marianna Alunni-Fabbroni, Sophia Kästle, Matthias Stechele, Jens Ricke, Michel Eisenblätter, Moritz Wildgruber

**Affiliations:** 1Department of Radiology, University Hospital, LMU Munich, 81377 Munich, Germany; melanie.kimm@med.uni-muenchen.de (M.A.K.); christopher.klenk@med.uni-muenchen.de (C.K.); Marianna.Alunni@med.uni-muennchen.de (M.A.-F.); sophia.kaestle@med.uni-muenchen.de (S.K.); matthias.stechele@med.uni-muenchen.de (M.S.); jens.ricke@med.uni-muenchen.de (J.R.); 2Department of Diagnostic and Interventional Radiology, Freiburg University Hospital, 79106 Freiburg, Germany; michel.eisenblaetter@uniklinik-freiburg.de

**Keywords:** tumor microenvironment, monocytes, macrophages, molecular imaging, polarization

## Abstract

Tumor-associated macrophages (TAMs) represent the largest group of leukocytes within the tumor microenvironment (TME) of solid tumors and orchestrate the composition of anti- as well as pro-tumorigenic factors. This makes TAMs an excellent target for novel cancer therapies. The plasticity of TAMs resulting in varying membrane receptors and expression of intracellular proteins allow the specific characterization of different subsets of TAMs. Those markers similarly allow tracking of TAMs by different means of molecular imaging. This review aims to provides an overview of the origin of tumor-associated macrophages, their polarization in different subtypes, and how characteristic markers of the subtypes can be used as targets for molecular imaging and theranostic approaches.

## 1. Formation of the Tumor Microenvironment

The formation of a pro-tumorigenic microenvironment is crucial for the survival of cancer cells at the primary and distant sites. The interplay between tumor cells, stroma, components of the extracellular matrix (including chemokines, cytokines, and hormones) and the immune system decides about the faith of primary tumors and metastases. Each stage of tumor development is characterized by its metabolic and cellular fingerprint influencing homeostasis between tumor progression and destruction.

The transformation of a normal cell into a tumor cell follows the accumulation of mutations in cancer genes. These mutations can be caused by errors during DNA replication or due to substantial DNA damage [1]. The accumulation of mutations leads to alterations of the cell and transforms it into a cancer cell. However, one cancer cell alone does not form a tumor. During evolution, the organism has developed several protection mechanisms for the elimination of cells that do not behave normally [2]. Thus, in order to expand, the cancer cells must escape this surveillance and build up their own tumor-friendly microenvironment [3].

Compared to a normal tissue environment, TME differs in architecture, nutritional state, pH value, oxygen levels, and the composition of the extracellular matrix. Remodeling of the extracellular matrix by cancer-associated fibroblasts (CAF) and stromal expansion results in a stiff environment with increased interstitial pressure [4]. This physical force together with biochemical signals facilitates epithelial-mesenchymal transition (EMT) as gene expression can be modulated [5,6,7]. Another hallmark of the TME is the creation of its own immune microenvironment, which encompasses T and B lymphocytes, NK cells, monocytes and myeloid-derived suppressor cells (MDSCs), mast cells, granulocytes, dendritic cells (DC), neutrophils, CAF, adipocytes, vascular endothelial cells, pericytes, and tumor-associated macrophages (TAMs) [8]. Furthermore, the metabolic reprogramming of the tumor environment (lactate and lipid metabolism, reactive oxygen species (ROS)) and the creation of specific conditions such as hypoxia is an essential aspect [9,10,11]. Tumor-derived secreted factors, chemokines, cytokines, and exosomes complete the TME [12,13,14,15,16]. During tumor growth, the TME constantly develops further and at any time point encompasses a heterogeneous landscape [17,18]. This heterogeneity hampers the effect of highly specific therapies. Thus, revealing the patient-specific microenvironment at the primary tumor level and at the distant metastatic site will allow for a better prediction of therapy response and potentially enable consecutive tailoring of therapy regimen [18,19,20]. The TME in this context provides targets for novel therapies aiming at normalization of the microenvironment to facilitate successful treatment of cancer patients [21,22,23]. In general, a healthy environment inhibits tumorigenesis, and thus, converting specific TME characteristics back to non-tumorigenic environment could be the key to improve cancer therapy.

## 2. Myeloid Cells: Monocytes and M-MDSCs

At every stage of disease progression, the tumor cells exert influence on the associated microenvironment, and in turn, the TME influences the tumor cells.

Chronic inflammation is an accepted hallmark of cancer, and inflammatory factors can both promote and eliminate tumor cells [24,25]. The current understanding of the complex role of inflammatory cells in the TME and their contribution to tumorigenesis is the following: at early stages of tumor development, cytotoxic immune cells (CD8^+^ T cells and NK cells) recognize and eliminate immunogenic tumor cells as a physiological mechanism [26,27]. However, remaining less immunogenic tumor cells become invisible to the immune cells and are thus able to expand. In addition, by downregulating tumor antigens and NK activation factors on the surface of tumor cells, they are able to escape their recognition and elimination by cytotoxic T cells and NK cells [2,28,29]. Moreover, tumor-derived, immunosuppressive signaling molecules such as IL-10, TGF-β, VEGF, and prostaglandin E2, as well as the expression of checkpoint molecules such as PD-L1 and CTLA-4, pave the way for the recruitment of MDSCs, TAMs, and T_reg_, orchestrated by tumor cell-derived chemokines such as CCL2, CSF1, CCL5, CCL22, CXCL5, CXCL8, and CXCL12 [15,30,31,32,33,34,35,36,37].

Monocytes develop from myeloid progenitor cells in the bone marrow, enter the circulation and migrate into various tissues where they differentiate to macrophages and DCs [38,39]. As monocytes, macrophages and DCs are capable of regulating T cell responses, therefore bridging innate and adaptive immunity [40]. Monocytes comprise a heterogenous group and were originally divided into a classical (CD16^-^CD14^+^), a non-classical (CD16^+^CD14^low^), and an intermediate (CD16^+^CD14^+^) subtype [41,42]. They are characterized by high functional plasticity, including the production of cytokines, clearance of pathogens, and antigen presentation. It has been shown that classical monocytes convert into non-classical over time [43,44]. How long each of the subtypes remains within the circulation and their relationship among each other is still under investigation. Only recently, the classification of monocytes was replaced in favor of their function (inflammatory, patrolling, and immunosuppressive monocytes) rather than classifying solely with respect to cell surface profiles [42,45,46,47,48]. The nomenclature of monocyte subsets is still not harmonized, and both classification types can be found. Inflammatory monocytes represent the classical subtype characterized by the expression of the CCR2 chemokine receptor, necessary to egress from the bone marrow and enter the blood circulation. At steady-state, inflammatory monocytes invade tissues and fill up the pool of tissue-resident monocyte-derived cells, finally differentiating into macrophages and dendritic cells. In addition, inflammatory monocytes are able to remain in their monocyte-like state and reside as local monocytes within the tissue [49]. Patrolling monocytes are the guardians of the vasculature and equal the non-classical monocyte subtype [50]. They are more differentiated than classical monocytes and characterized by the expression of CX3CR1. They survey the endothelium by constantly crawling along the lumen of the vasculature in an LFA/ICAM-dependent matter [46]. Studies in mice have shown that patrolling monocytes might not only descend from inflammatory monocytes but may also originate directly from a monocyte progenitor cell [51,52]. The intermediate monocyte subset (which is absent in mice) is hypothesized to be monocytes in transition [43], which could explain why there is no functional nomenclature for this subtype so far. At steady-state, they exhibit phagocytic and anti-inflammatory functions as well as high levels of intracellular IL-1β and TNF-α (pro-inflammatory mediators). The last subtype of functional monocytes is the immunosuppressive one including the monocytic myeloid-derived suppressor cells (M-MDSCs) which are mainly considered separately from the other monocyte subsets. They are pathologically activated myeloid cells and involved in facilitating tumor escape [53]. Their surface marker profile CD11b^+^CD14^+^CD15^-^HLA-DR^low/−^ allows the distinction between M-MDSCs and inflammatory and patrolling monocytes. Yet, under non-pathological conditions, the immature myeloid cells differentiate into monocytes. But under pathological conditions (stress, inflammation, and cancer) immature myeloid cells respond to emergency signals from the pathological site, preventing them from full differentiation and turning them into M-MDSCs [54]. Compared to monocytes, M-MDSCs exhibit differences in their gene expression and biochemical profile, which relates to their different functional activity. For separating monocytes and MDSC populations, HLA-DR is the most common marker as monocytes are positive for HLA-DR, whereas MDSCs are either low expressors or negative [55]. The immunosuppressive activity of M-MDSC is attributed to the secretion of arginase 1 (ARG1), inducible nitric oxide synthase (iNOS), as well as the production of ROS and nitric oxide (NO). They further express immune checkpoint molecules such as PD-L1 and CTLA-4 [55,56]. M-MDSCs specifically inhibit effector T cells within the TME in an antigen-specific and non-specific manner [56,57]. It is believed that M-MDSCs have evolved as protection from uncontrolled immune response associated with unresolved inflammation and M-MDSCs are a common feature of sepsis-induced immunoparalysis [58,59]. As a result, M-MDSCs in sepsis lead to the exhaustion of lymphocytes, which in turn stop the production of cytokines such as IFN-γ and IL-12. This mechanism also supports tumor cell survival, and thus, therapeutic targeting of M-MDSCs within the TME is a promissing tool to convert the tumor microenvironment into a normal tissue microenvironment and thus control cancer growth.

## 3. Recruitment and Polarization of M-MDSCs and Macrophages

All tumors exhibit a complex landscape of myeloid cells which are, among other cells, responsible for therapeutic responses [60]. Tumor cells and tumor-associated fibroblasts drive cancer-associated myelopoiesis by secreting soluble factors leading to the recruitment and expansion of myeloid cells from the bone marrow. These signals not only increase the myeloid output from the bone marrow but also modify the hematopoietic niche and alter myelopoiesis [61]. Among them are chemokines such as CCL2, colony-stimulating factors (G-CSF, M-CSF), stem cell factors, VEGF, cytokines such as IL-3 and IL-6, and regulatory proteins such as S100A8 and S100A9 [62,63,64,65,66,67,68]. In addition, these stimuli prevent immature myeloid cells from differentiating into monocytes leading to the formation of M-MDSCs. The pathological activation of M-MDSC already starts in the bone marrow mainly by inflammatory cytokines, which also attract M-MDSCs to the tumor site. Here, M-MDSCs either differentiate into tumor-associated macrophages (TAMs) or remain as M-MDSC population in the TME, where they play an important role in tumor progression as they directly act on the function of other immune cells. Per definition, M-MDSCs develop upon pathological conditions and exhibit immunosuppressive functions. Depending on the growth factors and inflammatory mediators in the environment, MDSCs differentiate in M1- or M2-like cells (according to the classification of T helper cells and macrophages), which can be identified by specific molecular markers [69,70] and functions (Figure 1). M1-like M-MDSCs have been shown to suppress tumor growth by increasing the amounts of free radicals (NO), death ligands (TNF-α), and immune cell-stimulating cytokine IL-12 [71,72]. In tumor tissue, mainly pro-tumorigenic, M2-like M-MDSCs are the dominant type, which inhibit tumor cell killing mediated by cytotoxic T lymphocytes primarily through IL-10, TGF-β, and ARG1 [73]. Furthermore, several studies have shown the importance of toll-like receptors (TLR) and IFNγ for M1-like M-MDSCs and IL-4 and IL-13 for the M2-like type (Table 1). So far, several compounds have been identified that are able to shift M2 M-MDSCs to M1 M-MDSCs, opening a new field to treat cancer [74]. Downstream pathways of potential targets in the reprograming process need to be investigated, but as MDSCs are crucial players in regulating the immune response in tumors. Understanding their molecular signature and behavior may be a game-changer in the treatment of a variety of diseases. Next to their own function as M1 or M2 M-MDSCs, studies have shown that within tumor tissue, they are able to develop further into TAMs. Thus, they comprise a class of monocytes which, in addition to classical monocytes (inflammatory and patrolling), can give rise to macrophages within tumor tissue [75].

This finding illustrates why macrophages also display such a heterogeneous group. Furthermore, factors in the tissue environment determine their function. Macrophages are terminally differentiated myeloid cells that can be found in all tissues. Some of them derive from circulating monocytes, some of them from tissue-resident monocytes, and some of them from tissue-resident macrophages [76]. The ontogeny of TAMs is still under investigation, but analogous to other immune cells, two subtypes of TAMs, M1 (anti-tumorigenic) and M2 (pro-tumorigenic) phenotypes are described [77,78] (Figure 1).

To make it even more complicated, there is more than one pool where TAMs are recruited from. The study from Fogg et al. [79] identified a precursor for monocytes, macrophages, and dendritic cells and emphasized the potential that monocytes can reconstitute tissue macrophages under certain conditions. A monocyte-restricted precursor was found in the bone marrow, which derived from a common progenitor and is distinguishable to a common dendritic precursor by differences in *c-Kit* expression and loss of the CD135 surface marker [80]. Even though several studies showed that circulating monocytes can give rise to tissue-resident macrophages, it is not yet fully clarified to what extent they do differentiate into macrophages. Over time, more and more details about tissue-resident macrophages have been identified and finally lead to today’s view on how macrophages derive. For example, macrophages in the brain (microglia) derive from the yolk sac. Tissue-resident macrophages from other organs such as the gut, heart, lung (alveolar macrophages), spleen, bone (osteoclasts), skin (Langerhans cells), and liver (Kupffer cells) comprise cells of multiple developmental origins, all of them being established prior to birth and able to maintain themselves independently from blood monocytes during adulthood [81,82]. The number of tissue-resident macrophages seems to play a crucial physiological role, and changes within the population correlate with early aging [83]. A certain percentage of tissue-resident macrophages is continuously updated from blood-derived myeloid cells but differs from tissue to tissue and changes upon disease [84]. As an example, microglia strictly repopulate from yolk sac-derived progenitor cells, whereas heart-resident macrophages are replaced by blood-derived monocytes [85,86]. Upon injury, Kupffer cells in the liver can derive from liver-resident macrophages, monocyte-derived macrophages, or peritoneal macrophages [87]. The latest results from single-cell RNA sequencing of human liver macrophages revealed the existence of two distinct populations next to each other, one of which is anti-inflammatory, the other pro-inflammatory [88]. The ontogeny of these two populations still needs to be elucidated, and it could very well be that the pro-inflammatory subtype derives from circulating monocytes. At steady-state, Kupffer cells renew independently from bone marrow-derived progenitors, but recent studies showed that monocyte-derived macrophages might be an alternative source for Kupffer cells under certain conditions. The activation of chemokine ligands (e.g., CCL2, CXCL1, and CXCL10) on Kupffer or hepatic stellate cells seems to be the main driver for monocyte infiltration into the liver following injury or infection [89,90,91,92]. The least well-studied population of liver-resident macrophages are the peritoneal macrophages. Wang et al. were able to identify F4/80^hi^GATA6^+^CD11b^+^ mature macrophages to accumulate in the liver following thermal injury [93]. They further reported that none of the bone marrow-derived monocyte subpopulations infiltrated the site of injury. When it comes to injury, spleen-derived monocytes also enter the game [94]. During the development of liver fibrosis, splenic macrophages fuel the hepatic inflammation by secreting pro-inflammatory mediators into the portal vein [95,96]. Hematopoietic stem and progenitor cells (HSPCs) found in the spleen were phenotypically and functionally analogous to HSPCs found in the bone marrow and they also gave rise to monocytes that were able to enter the tumor [97,98]. Studies on the influence of splenectomy revealed a role of the spleen in immune surveillance and protection from tumor development and metastasis [99,100].

Summing it up, macrophages in tumor tissue most likely present a heterogeneous population of cells derived from monocytes from bone marrow, spleen, and blood as well as tissue-resident macrophages. The fate of TAMs (M1 or M2 subtype) might be pre-determined by the origin of the cell and by tumor-derived mediators present in the blood and in the tumor microenvironment (Figure 2).

## 4. TAM Targeting

Once macrophages are located in the TME, their function depends on the subtype: pro-inflammatory M1-like TAMs show a tumor-suppressive behavior, promoting a positive therapeutic response and improved patient survival. In contrast, M2-like TAMs promote primary tumor survival, metastasis, and correlate with a worse therapeutic outcome. As M2-like TAMs are the predominant population of macrophages within the tumor tissue, TAMs and M2-activated macrophages (in tumor tissue) are often used synonymously in the literature. Even though M1- and M2-like TAMs harbor distinct profiles, TAMs are regarded as remarkable plastic, and the subtypes can pass over from one to the other [101]. Initially, macrophages were classified according to their activation agent: LPS and IFN-γ activated M1 type and IL-4 and IL-10 activated M2 type [81]. Nowadays, macrophage grouping is primarily related to the function of the macrophage following the monocyte classification. Even though M1 and M2 TAMs show different genetic profiles, the favored idea is that TAMs differ in their function rather than their phenotype [101]. This idea is supported by studies showing that various stimuli lead to an alteration in the TAM phenotype allowing a switch from M1 into M2 and vice versa [102,103]. In particular, the repolarization of TAMs into the pro-inflammatory M1-like type is the subject of numerous investigations. The microenvironment is the most crucial factor for TAM polarization [104], which makes it even harder to explain why M1- and M2-like TAMs together with intermediate forms, expressing genes from both types, do exist next to each other [105]. Recently, subtypes of M2-like TAMs have been identified and classified as M2a, M2b, M2c, and others [106,107]. The induction of one or the other subtype is a result of different stimuli in the environment. M2a macrophages express high levels of mannose receptors, M2b macrophages are known as regulatory ones, and M2c macrophages are induced by IL-10 [108,109,110]. The proper characterization of these subtypes is important in regard to tumor therapy as specific targeting reduces side effects and improves the reformation of the microenvironment toward a tumor-suppressive one.

Next to the polarization of TAMs by chemokines and cytokines, the interaction with tumor-derived extracellular vesicles (EVs) plays an important role in TAM determination. In addition, several micro RNAs (miRNA or MiR) have been described to be involved in the differentiation of myeloid cells into macrophages and also in the polarization of TAMs [111,112,113]. M2-like TAMs overexpressing miRNA-155 repolarize to M1 [114], and inhibition of miRNA-155 impairs M1 functions. M1 polarization is also assisted by miRNA-127 and miRNA-125b [115,116] (Table 1). EVs and exosomes carrying RNAs and proteins from M1-like TAMs are capable of switching M2-like TAMs to M1 and potentiate an anti-tumor effect. As exosomes can act as a carrier to deliver therapeutics into the tumor [117], several efforts have been made to diminish TAMs directly or to repolarize them by using EVs as a vehicle. Another strategy is to block monocyte infiltration in tumor tissue, cutting down one resource of TAMs. CCR2 expressing monocytes are driving tumor progression, and therefore, inhibiting the CCL2-CCR2 axis could stop tumor expansion [118]. Several clinical trials with CCR2 blockers are ongoing [119]. Targeting CXCR4 has also shown promising results in preclinical models [120]. The direct depletion of TAMs is another strategy to support anti-tumor therapies. Bisphosphonates such as zoledronate, clodronate, and trabectedin have been shown to reduce TAM survival by inducting cell cycle arrest [121]. However, depletion of TAMs might have the consequence that all myeloid cells in the tumor are getting destroyed, and the loss of pro-inflammatory macrophages could cause side effects and even fire tumor progression as macrophages are needed for efficient immunotherapy [122]. Thus, repolarization of M2-like TAMs is assumed to be the better strategy. The most effective way seems to include targeting both the tumor cell (e.g., CD47/SIRPα) and TAMs (e.g., TLR antagonist, PI3Kγ inhibitor, CD40 agonist, and HDAC inhibitors) [123,124]. Loading drugs into macrophages can be accomplished using nanoparticles as carriers [125]. Silica and gold nanoparticles, liposomes, and other polymers are currently under investigation. Very recently, live cells are executed to deliver drugs to tumor cells. Monocytes/macrophages are ideal candidates as they can ingest relatively large amounts of drugs, and they are easily recruited to tumors [126,127]. Preclinical and clinical studies targeting TAMs are listed in Table 2.

**Table 1 biomedicines-09-00374-t001:** Common macrophage markers.

M1		M2
CD80, CD86 [128,129]	CD surface receptor	CD163, CD206, CD200R [128,130,131]
CXCL8, CXCL9, CXCL10, CCL2, CCL3, CCL5 [132]	Chemokines	CXCL12, CCL2,3,4,5,18,20 [133,134]
IL-1β, IL-2, IL-6, IL-12, IL-23, IFN-γ, TNF-α [135,136,137]	Cytokines	IL-4, IL-6, IL-10, IL-13, TGF-β, EGF [130,138,139,140,141,142,143,144,145,146]
MHC class II [147,148]	Biomarker	S100A8, S100A9, MMP2, MMP9, STAB1 [149,150,151,152,153]
	Vasculature marker	VEGF [154,155]
STAT1, IRF3, IRF5, HIF1α, AP1 [156,157,158,159]	Transcription Factors	STAT3, IRF4, FIZZ1, YM1 [160,161,162,163]
	Checkpoint proteins	PD-L1 [164,165]
iNOS, NO, ROS, IDO, PFKFB3, PKM2, ACOD1 [166,167,168,169,170,171]	Metabolites	ARG1, IDO, CARKL, GS [140,149,171,172,173,174,175]
miR-9, miR-18, miR-19a/b, miR-21, miR-26, miR-27a/b, miR-29b, miR-33, miR-125b, miR-127, miR-130a, miR-143, miR-145, miR-147, miR-155, miR-216a, miR-330 [176,177,178,179]	miRNA	miR-21, miR-23a/b, miR-24, miR-27a, miR-29, miR-34a, miR-124, miR-125a, miR-132, miR-146a, miR-155, miR-181, miR-188, miR-223, miR-511 [176,179,180,181,182]

## 5. TAM Tracking and Imaging

Imaging of TAMs and tracking their fate over time is not only necessary to understand the dynamic role of mononuclear phagocytes within the TME but also to monitor therapies that are targeting TAMs specifically. As TAMs are the most abundant cells within the TME, imaging can be performed with higher signal-to-noise compared to less abundant immune cells in the TME. Molecular sensors can target TAMs in principle by three different means: (i) TAM-specific antibodies (such as CD68 or CD206) coupled to signal moieties of a different kind (magnetic resonance agents, radionuclides, or nanoparticles); (ii) the phagocytic or metabolic capacity can be used to introduce signal moieties into TAMs themselves; and (iii) reporter gene imaging of TAMs.

Targeting subset-specific receptors expressed by different TAMs may become suitable to assess the dominating polarization type abundant in the TME. Targeting the folate (M1) and the mannose receptor (M2) using dedicated near infrared (NIR) sensors or radiotracers has been explored primarily beyond the tumor field but can potentially be applied to assess the status and amplitude of immune activation in vivo [203,204,205]. Folate-conjugated fluorescein isothiocyanate (folate-FITC) was shown to capture tumor-infiltrating immune cells in a mouse model of head and neck squamous carcinoma [206]. Recently, folate-conjugated nanobubbles were evaluated for both TAM targeting and reeducation [207]. A nanobody-based targeting CD206 has been shown to selectively capture pro-angiogenic TAMs residing in hypoxic areas of the TME [208]. Using nanobodies, which are single-domain antigen-binding fragments derived from Camelidae heavy-chain antibodies, has the benefit of improved delivery and accumulation within the tumor, which is a hurdle of many large-molecule contrast agents and sensors. Antibodies, due to their size, frequently suffer from a poor penetration of solid tumors and a high Fc-mediated unspecific binding, thereby causing a poor signal-to-noise of the TME on in vivo imaging. Nanobodies instead are chemically stable, soluble, and exert a high affinity and increased tissue penetration, making them favorable for targeting the TME [209]. Coupling nanobodies with signal-giving moieties may thus overcome the current limitations of antibody-based molecular imaging, frequently suffering from an insufficient tumor-to-noise ratio. Labeling nanobodies targeting CD206 with ^99m^Tc has been shown to successfully image specific TAM subpopulations in vivo [208].

The simplest way to image macrophages in vivo is using their high phagocytic activity to take up nanoparticles. Labeling nanoparticles with various PET tracers for whole-body imaging can be either achieved through incorporating suitable isotopes or by chelation of such isotopes with the nanoparticle [210]. Locke et al. developed a mannosylated liposome loaded with ^64^Cu that accumulated in TAMs in a mouse model of pulmonary adenocarcinoma and was able to quantify the tumoral TAM load by PET imaging [211]. Of note, such a nanoparticle delivery system can not only be used for molecular imaging of TAM content and activity but can similarly be combined with delivering therapeutic agents to the TME in a theranostic approach [212,213,214]. As a general limitation, one has to mention that rarely are these approaches 100% specific, neither for TAMs nor a specific subset. For example, the mannose receptor is not exclusively expressed by TAMs with M2 polarization but also on resident macrophages and other phagocytes. As TAMs were identified as the main containers for high-density lipoproteins (HDLs) with the TME, using the HDLs as delivery cargo for nanoparticle-based tracers may increase the specificity of TAM imaging. Selective targeting of TAMs was observed in an orthotopic murine breast cancer model by using ^89^Zr-reconstituted HDL as a PET tracer [215]. Besides addressing the mere presence of TAMs, tracking their biological activity may be even more meaningful, especially when monitoring therapeutic effects of TME specific therapies. S100A8/A9 has been identified as a promising marker for monocyte activation within the inflammatory TME [216]. Fluorescence imaging of the heterodimer protein S100A8/A9 has been shown to report specific features of malignancy in a murine breast cancer model [217,218] (Figure 3). Using a similar model, Eisenblätter et al. showed that S100A8/A9 can be used as an imaging biomarker for pre-metastatic tissue priming. S100A8/A9 expression detected by SPECT revealed myeloid-derived suppressor cell abundance in pre-metastatic lung tissue before the onset of actual metastasis. S100A8/A9 signal correlated with subsequent tumor burden of lung metastasis and selective CCL2 blockade was able to modulate TAM activity within the TME and influence subsequent metastatic growth [219]. S100A9 NIR imaging further allowed to capture therapy-mediated changes of the inflammatory tumor microenvironment following Doxorubicin/Bevacizumab treatment or murine breast cancer xenografts [217].

The most established way of tracking macrophages is labeling with small or ultrasmall superparamagnetic iron oxide nanoparticles (SPIO/USPIO). Those particles are preferentially taken up by monocytes/macrophages via phagocytosis and induce a strong signal decay on T2* weighted magnetic resonance imaging [221,222]. Tracking of TAM accumulation within the TME during tumor growth has been initially reported by Shih et al. [223]. It is important to note that in mice and humans, phagocytosis activity may differ between distinct monocyte/macrophage subsets [221,224]. A distinct design of magneto-fluorescent nanoparticles can render the particle specific for certain macrophage polarizations [224]. The application of SPIO for TAM imaging has been extensively reviewed elsewhere [225]. The advantage of the SPIO approach is that there are particles that have been clinically approved for liver MR imaging carrying a very low toxic potential. In a first-in-patient trial of ferumoxytol, enhanced MRI showed that MR signal enhancement correlated with the presence of CD68^+^CD163^+^ TAMs [226]. Intriguing was the report that clinically approved SPIOs not only enable imaging TAMs in vivo non-invasively but that they similarly inhibit tumor growth by triggering a pro-inflammatory TAM polarization with the TME. The authors conclude that ferumoxytol thereby might protect subsequent organs from metastatic seeds or potentiate macrophage-modulating cancer immunotherapies [227].

TAM tracking with SPIO by MR imaging similarly underlies some major limitations. First, direct (ex vivo) as well as in vivo labeling of monocyte/macrophages with iron oxide nanoparticles is non-specific to TAMs. Many other cells, such as neutrophils, carry some (although reduced) phagocytic potential. Especially organs with a physiologically high amount of resident phagocytic cells such as the liver (Kupfer cells) limit the specificity of such TAM imaging approaches. Second, image signal quantification of T2* MRI is highly complex and incomplete as frequently there is a non-linear correlation between local nanoparticle concentration and the obtained imaging signal. This lack of quantification can be improved by adding ex vivo mass spectrometry imaging such as laser ablation inductively coupled plasma mass spectrometry to in-vivo MRI. Applying a customized ^57^Fe based SPIO (instead of using 56Fe, which is highly abundant in hemoglobin, myoglobin etc.) Masthoff and Faber showed that spatially resolved maps of local nanoparticle distribution can be used to correct the in vivo imaging results, thereby improving quantification [228,229]. Third, noninvasive whole-body imaging is confined to cell tracking and assessing the gross magnitude of immune response in cancer. However, until now, nanoparticle MRI is not able to depict interactions of TAMs with other components of the TME. These cellular and subcellular processes can only be investigated by means of intravital microscopy, such as confocal or multi-photon approaches. Although the spatial resolution of MRI in principle would be sufficient to depict molecular processes at cellular resolution, the temporal resolution until now is not able to capture immune cell dynamics in real-time. Novel time-lapse MRI, by creating video-like loops of SPIO-labeled immune cells within the circulation, may overcome this limitation in the future and capture moving monocytes in situ [230]. Table 3 is summarizing the actual targets and probes used for TAM imaging.

Restricted to the preclinical setting, reporter gene imaging is additionally promising to assess TAM presence and activity in vivo. Activated luciferase-expressing macrophages created using the Cre-Lox system could be tracked in murine melanoma and breast cancer models [244]. Choi et al. were able to visualize the migration of TAMs to solid tumor lesions and evaluated the effect of anti-inflammatory therapy on the TME in a murine colon cancer model [245]. Putting a luciferase reporter under the control of the Arg1 promotor, indicative of M2 polarization, enabled bioluminescence tracking of adopted macrophages migrating toward the TME, thus allowing to investigate of even TAM polarization in vivo by gene reporter imaging [246].

## 6. Perspective

Macrophages represent the largest pool of immune cells in solid tumors, and different types of macrophages with divergent functions were identified so far. Depending on their profile and function, they are categorized into tumor-promoting or tumor-suppressive types. Results from several studies suggest that during early tumor manifestation and expansion, M1-type macrophages are the more dominant type. At late tumor stages, especially when metastases have already occurred, M2-type macrophages take over the scene. Several efforts are made to target tumor-associated macrophages by means of molecular imaging. Various methods exist that are able to track tumor-associated macrophages non-invasively. Specificity of the molecular probe toward a dedicated M1/M2 target, as well as a high sensitivity of the signal moiety of the imaging agent, is decisive for successful imaging of TAM dynamics in vivo. Molecular imaging thereby can help to better understand the time frames in which TAMs are specifically important during tumor growth and thus identify the optimal time points for TAM-directed therapies. Thereby, molecular TAM imaging can play a significant role within the precision medicine conundrum: to treat the right patient with the right drug at the right time point. Additionally, methodologies from molecular TAM imaging can be used in theranostic approaches, where imaging and reprogramming of TAMs can occur at the same time.

## Figures and Tables

**Figure 1 biomedicines-09-00374-f001:**
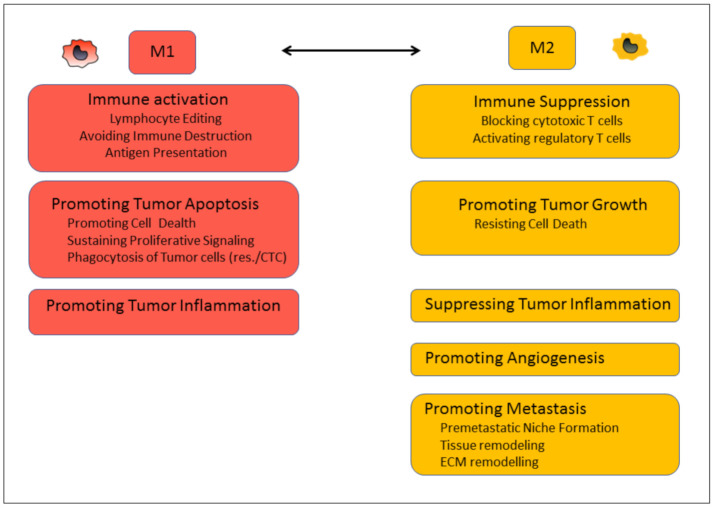
Functional hallmarks of M1 and M2 TAMs.

**Figure 2 biomedicines-09-00374-f002:**
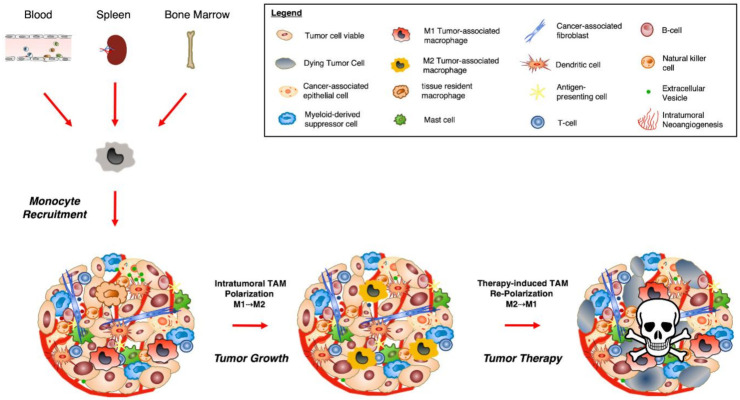
Monocyte recruitment and TAM polarization during tumor development and therapy. Monocytes from different compartments (bone marrow, blood, spleen) are directed to the tumor microenvironment (TME) via secreted factors and exosomes. Here, they differentiate into TAMs of different subtypes (M1/M2). Therapies leading to the repolarization of M2 TAMs into M1 TAMs or monocytes are promising strategies for tumor destruction.

**Figure 3 biomedicines-09-00374-f003:**
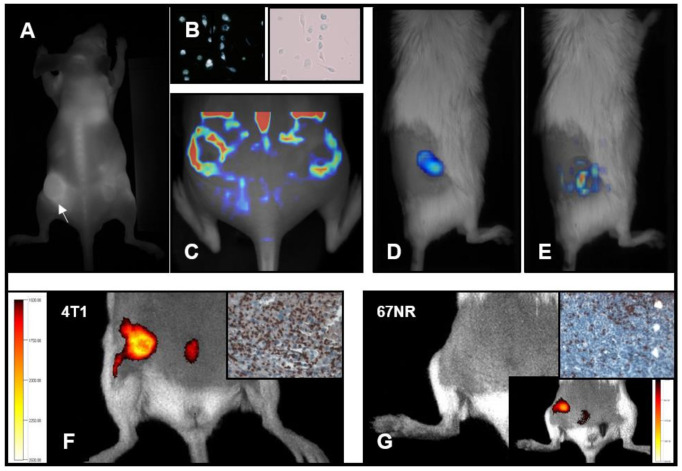
TAM tracking using optical imaging. (**A**–**C**) DiR-labeled macrophages (**B**), directly tracked 48 h after intravenous application using FRI (**A**) and FMT (**C**). The signal represents local macrophage accumulation in 4T1 tumors, implanted in the flanks of mice (adopted with permission of [220]). (**D**–**G**) Homogeneous tracer distribution in tumors after injection of control IgG-Cy7 (**D**) and TAM-targeted S100A9-Cy5.5, which accumulates in the tumor periphery (**E**). The TAM activity reflects the tumor malignancy: S100A9-Cy5.5 gives a stronger signal in highly malignant 4T1 tumors (**F**) than in slow-growing, less aggressive 67NR tumors (**G**). Inserts (upper right corner) show corresponding immunohistochemistry of S100A9^+^ TAMs (adopted with permission of [218]).

**Table 2 biomedicines-09-00374-t002:** TAM targeting therapies in preclinical or clinical studies.

TAM Recruitment	TAM Depletion	TAM Reprogramming
CCCR2-CCL2 inhibition [183,184,185]	Trabectedin [186,187]	Class IIa HDAC inhibitors [188,189]
CXCR4-CXCL12 inhibition [190]	Biphosphonates [121,191]	CD40 agonists [192,193]
	Anti-CSF-1R [194,195]	PI3Kγ inhibitors [196]
		SIRPα inhibitors [197,198]
		STAB1 inhibitors [199]
		Checkpoint inhibitors [200]
		TLRs agonists [201]
		siRNA/miRNA [202]

**Table 3 biomedicines-09-00374-t003:** Target molecules and myeloid-specific processes used for TAM-specific imaging.

Target Process/Molecule	Modality	Tracer
Phagocytosis	MRI, PET, optic, hybrid	Iron oxide nanoparticles [231], ^64^Cu-labeled polyglucose nanoparticle (macrin) [232], Cy5.5-VEGF [233], perfluorocarbon (PFC) [234]
Endocytosis	PET	^89^Zr-PL-HDL, ^89^Zr-AI-HDL [215]
F4/80	Flow cytometry, fluorescence microscopy	^111^In-αF4/80-A3-1 mAb [235]
CD11b	PET	^18^F-VHHDC13 [236]
MHC-II	PET	^18^F-VHH7 [236]
CD206	PET	γ-Tilmanocept [237], 18F-FB-anti-MMR 3.49 sdAb [238]
CD163	PET	^68^Ga-αCD163-mAb [239]
TSPO	PET	^18^F-GE-180 [240], ^18^F-DPA-714 [241], ^11^C-PBR28 [242]
FR-ß	SPECT, PET	3′-Aza-2′[^18^F]fluorofolic acid [243]

## Data Availability

Not applicable.

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
