# Peer review of "Tumor-Associated Macrophages—Implications for Molecular Oncology and Imaging"

_biomedicines, 2021, doi:10.3390/biomedicines9040374_

Round 1

Reviewer 1 Report

Necessary clarification, elaboration and additions:

  1. “Monocytes develop from myeloid progenitor cells in the bone marrow, enter the circulation and migrate into various tissues, differentiating into resident macrophages 77 [38,39].” This statement is inaccurate regarding the origin of resident macrophages in normal tissues.
  2. “Compared to monocytes, M-MDSC exhibit a different genetic and biochemical profile as well as a different functional activity.” Please elaborate on “genetic profile” of M-MDSC.
  3. “They further express immune checkpoint molecules like PD-L1 and CTLA-4 [55,56].” Original articles, rather than a review paper, for expression of CTLA-4 in M-MDSC should be cited.
  4. “Tumor cells and tumor-associated fibroblasts drive cancer-associated myelopoiesis by secreting emergency signals.” Please elaborate on the molecular natures of “emergency signals”
  5. Lacking information regarding intra-tumor distribution of different subtypes of M-MDSC. Fig. 2 could be more informative by incorporating recent discoveries regarding localization/distribution patterns of M-MDSC in tumors
  6. It will be helpful to have a table to summarize the information of clinical/preclinical TAM targeting therapies, such as specific targets at the levels of molecules, cell types and tumor types.
  7. It will be helpful to have a table to summarize the information of molecules targeted by current imaging technology, such as their specific expression in different subtypes of M-MDSC and limitation of their usage.

Author Response

Munich, 28th March 2021

RE: Manucscript revision "Tumor associated macrophages – Implications for molecular Oncology and Imaging" by Melanie A. Kimm and co-authors

Manuscript ID: biomedicines-1151111

Dear Mrs. Soceanu, dear members of the editorial board, honored reviewers,

we would like express our sincere gratitude for the careful revision of the manuscript. The points raised by the reviewers helped us to further improve our manuscript.

We considered each point by the reviewers and reworked the manuscript accordingly. We provide a detailed point-by-point reply below in response to the reviewers’ comments.

We hope that we could answer each point in a satisfactory manner and hope that the manuscript is now acceptable for publication in Biomedicines.

Sincerely yours,

Melanie A. Kimm and Moritz Wildgruber

(on behalf of the authors)

Point-by-point response

Academic Editor Comments:

In general this a good review, giving an update to the topic of tumor associated macrophages. However there is one point I would like to be improved. Data presented in Table 1 are given as something generally known, without providing citations to each and every marker. This might be fine in regard of really well described markers, although I personally would expect citations also there. But in the case of less known markers it is absolutely necessary.  For example IDO1 that is presented as M2 marker is activated by IFN-gamma, a known inducer of M1 phenotype (Adv Exp Med Biol. 2021;1275:339-356.  doi: 10.1007/978-3-030-49844-3_13., Oncol Rep. 2021 Jan;45(1):379-389.  doi: 10.3892/or.2020.7837.  Epub 2020 Nov 5.). As well CSF1 is induced both by IL-4 and IFN-gamma, so it is not a marker of M2. Some important M2 markers (like Stabilin-1) are missing.

We thank the academic editor for this encouraging comment. We have reworked the entire Table 1 and added references accordingly.

Table 1. Common Macrophage Markers

M1

M2

CD80, CD86 [128,129]

CD surface receptor

CD163, CD206, CD200R [128,130,131]

CXCL8, CXCL9, CXCL10, CCL2, CCL3, CCL5 [132]

Chemokines

CXCL12, CCL2,3,4,5,18,20 [133,134]

IL-1, IL-2, IL-6, IL-12, IL-23, IFN-, TNF- [135–137]

Cytokines

IL-4, IL-6, IL-10, IL-13, TGF-, EGF [130,138–146]

MHC class II [147,148]

Biomarker

S100A8, S100A9, MMP2, MMP9, STAB1 [149–153]

Vasculature marker

VEGF [154,155]

STAT1, IRF3, IRF5, HIF1, AP1 [156–159]

Transcription Factors

STAT3, IRF4, FIZZ1, YM1 [160–163]

Checkpoint proteins

PD-L1 [164,165]

iNOS, NO, ROS, IDO, PFKFB3, PKM2, ACOD1 [166–171]

Metabolites

ARG1, IDO, CARKL, GS [140,149,171–175]

miR-9, miR-18, miR-19a/b, miR-21, miR-26, miR-27a/b, miR-29b, miR-33, miR-125b, miR-127, miR-130a, miR-143, miR-145, miR-147, miR-155, miR-216a, miR-330 [176–179]

miRNA

miR-21, miR-23a/b, miR-24, miR-27a, miR-29, miR-34a,  miR-124, miR-125a, miR-132, miR-146a, miR-155, miR-181, miR-188,  miR-223, miR-511 [176,179–182]

Reviewer #1

Necessary clarification, elaboration and additions:

  1. “Monocytes develop from myeloid progenitor cells in the bone marrow, enter the circulation and migrate into various tissues, differentiating into resident macrophages 77 [38,39].” This statement is inaccurate regarding the origin of resident macrophages in normal tissues.

We thank the reviewer for the critical remark. The sentence was changed to the following (page 4, lines 95-97):

"Monocytes develop from myeloid progenitor cells in the bone marrow, enter the circulation and migrate into various tissues where they differentiate to macrophages and dendritic cells (DC) [38,39]."

  1. “Compared to monocytes, M-MDSC exhibit a different genetic and biochemical profile as well as a different functional activity.” Please elaborate on “genetic profile” of M-MDSC.

We thank the reviewer for the constructive remark regarding the genetic details of M-MDSC. To avoid misleading about the genetic nature of MDSC we have changed  and amended the sentence as follows (page 5, lines 136-139):

"Compared to monocytes, M-MDSC exhibit differences in their gene expression and biochemical profile which relates to their different functional activity. For separating monocytes and MDSC populations HLA-DR is the most common marker as monocytes are positive for HLA-DR whereas MDSC are low expressors or negative. [55]"  

  1. “They further express immune checkpoint molecules like PD-L1 and CTLA-4 [55,56].” Original articles, rather than a review paper, for expression of CTLA-4 in M-MDSC should be cited.

We apologize for the mistake and we have replaced the citation with original articles. The new original references are:

  1. Lu, C.; Redd, P.S.; Lee, J.R.; Savage, N.; Liu, K. The Expression Profiles and Regulation of PD-L1 in Tumor-Induced Myeloid-Derived Suppressor Cells. Oncoimmunology 2016, 5, e1247135, doi:10.1080/2162402x.2016.1247135.
  2. Clavijo, P.E.; Moore, E.C.; Chen, J.; Davis, R.J.; Friedman, J.; Kim, Y.; Waes, C.V.; Chen, Z.; Allen, C.T. Resistance to CTLA-4 Checkpoint Inhibition Reversed through Selective Elimination of Granulocytic Myeloid Cells. Oncotarget 2014, 5, 55804–55820, doi:10.18632/oncotarget.18437.

  1. “Tumor cells and tumor-associated fibroblasts drive cancer-associated myelopoiesis by secreting emergency signals.” Please elaborate on the molecular natures of “emergency signals”

We thank the reviewer for pointing this out. Indeed, the molecular nature of signals that lead to the recruitment, expansion and activation of MDSC should be explained in more detail. Therefore, we extended the manuscript as follows (page 6, lines 161-172):

"Tumor cells and tumor-associated fibroblasts drive cancer-associated myelopoiesis by secreting soluble factors leading to the recruitment and expansion of myeloid cells from the bone marrow. These signals not only increase the myeloid output from the bone marrow, but also modify the hematopoietic niche and alter myelopoiesis [60]. Amongst them are chemokines like CCL2, colony-stimulating factors (G-CSF, M-CSF), stem cell factors, VEGF, cytokines like IL-3 and IL-6 and regulatory proteins like S100A8 and S100A9 [61-67]. In addition, these stimuli prevent immature myeloid cells from differentiating into monocytes leading to the formation of M-MDSC. The pathological activation of M-MDSC already starts in the bone marrow mainly by inflammatory cytokines which also attract M-MDSC to the tumor site.  Here, M-MDSC either differentiate into tumor associated macrophages (TAMs) or remain as M-MDSC population in the TME where they play an important role in tumor progression as they directly act on the function of other immune cells."

  1. Lacking information regarding intra-tumor distribution of different subtypes of M-MDSC. Fig. 2 could be more informative by incorporating recent discoveries regarding localization/distribution patterns of M-MDSC in tumors

We thank the reviewer for this advice. The Figure with the different tumoral components is thought as a schematic overview showing that tumors are composed of much more than mere tumor cells and thereby including major players of the TME. The figure is therefore not indicative for the exact intratumoral distribution pattern and localization of certain components of the TME.  We have modified the figure to make the entire graphics more clear. Please see also response to Reviewer 2 point 2.

  1. It will be helpful to have a table to summarize the information of clinical/preclinical TAM targeting therapies, such as specific targets at the levels of molecules, cell types and tumor types.

We thank the reviewer for the advice. Please find the new Table 2 attached (page 16, lines 432-434):

Table 2. Target Molecules and myeloid specific processes used for TAM specific Imaging

Target process/molecule

Modality

Tracer

Phagocytosis

MRI, PET, Optic, Hybrid

Iron oxide nanoparticles [211], 64Cu-labeled polyglucose nanoparticle (Macrin) [212], Cy5.5-VEGF [213], Perfluorocarbon (PFC) [214]

Endocytosis

PET

89Zr-PL-HDL, 89Zr-AI-HDL [195]

F4/80

Flow cytometry, Fluorescence microscopy

111In-αF4/80-A3-1 mAb [215]

CD11b

PET

18F-VHHDC13 [216]

MHC-II

PET

18F-VHH7 [216]

CD206

PET

γ-Tilmanocept [217], 18F-FB-anti-MMR 3.49 sdAb [218]

CD163

PET

68Ga-αCD163-mAb [219]

TSPO

PET

18F-GE-180 [220], 18F-DPA-714 [221], 11C-PBR28 [222]

FR-ß

SPECT, PET

3’-Aza-2’[18F]fluorofolic acid [223]

  1. It will be helpful to have a table to summarize the information of molecules targeted by current imaging technology, such as their specific expression in different subtypes of M-MDSC and limitation of their usage.

We thank the reviewer for the remark and have added a new Table 3 to the manuscript as requested (pages 16-17, lines 445-446):

Table 3. TAM targeting therapies in preclinical or clinical studies

TAM Recruitment

TAM depletion

TAM reprogramming

CCCR2-CCL2 inhibition [227–229]

Trabectedin [230,231]

Class IIa HDAC inhibitors [232,233]

CXCR4-CXCL12 inhibition [234]

Biphosphonates [121,235]

CD40 agonists [236,237]

Anti-CSF-1R [238,239]

PI3Kγ inhibitors [240]

SIRP inhibitors [241,242]

STAB1 inhibitors [243]

Checkpoint inhibitors [244]

TLRs agonists [245]

siRNA/miRNA [246]

Reviewer 2 Report

Kimm et al. have made a good and valuable review of the state of the art regarding TAMs and their implications in tumor origins and progression, as well as cancer diagnosis and therapy. The article highlights important data, is very well structured, language is clear, the bibliography is up-to-date, and the different assessments are easy to follow. As a negative point, I found some of the ilustrations uninformative and poorly implemented. Nevertheless, and in general,  I recommend the manuscript for publication.

Minor revision. Only 3 points need some improvements:

  1. Page 5, lanes 189-190: This sentence needs to be more clear.
  2. Page 5, figure 2: Figure not informative, a lot of elements mixed with no sense. Font size of the legend too small and low image quality.
  3. Pages 10-16, References: The numbering is repeated twice.

Author Response

Reviewer #2

Kimm et al. have made a good and valuable review of the state of the art regarding TAMs and their implications in tumor origins and progression, as well as cancer diagnosis and therapy. The article highlights important data, is very well structured, language is clear, the bibliography is up-to-date, and the different assessments are easy to follow. As a negative point, I found some of the ilustrations uninformative and poorly implemented. Nevertheless, and in general,  I recommend the manuscript for publication.

We thank the reviewer for this encouraging comment. We have reworked the manuscript and corrected all points that have been raised.

Minor revision. Only 3 points need some improvements:

  1. Page 5, lanes 189-190: This sentence needs to be more clear.

We thank the reviewer for this helpful suggestion. We have modified the sentence as follows (pages 5-6, lines 146-157):

"As a result, M-MDSCs in sepsis lead to the exhaustion of lymphocytes which in turn stop the production of cytokines such as IFN- and IL-12. This mechanism also supports tumor cell survival and thus therapeutic targeting of M-MDSC within the TME is a hopeful candidate to convert tumor microenvironment into normal tissue microenvironment and thus control cancer growth."

  1. Page 5, figure 2: Figure not informative, a lot of elements mixed with no sense. Font size of the legend too small and low image quality.

We thank the reviewer for this constructive suggestion. The monocyte recruitment and TAM polarization is responding to cells and soluble factors in the blood and tissue microenvironment. As the microenvironment consists of multiple cell types, we aimed to represent the microenvironment as a major part of the tumor. The idea here is simply to show that the microenvironment is made up of multiple different components beyond the cancer cells themselves, many of them which actually interact with TAMs. We have revised Figure 2 to make it more easily readable (and with 300 dpi resolution). However, if the reviewers and editors feel the manuscript would be better with the figure entirely deleted, we would agree to this action.

  1. Pages 10-16, References: The numbering is repeated twice.

We have checked again the references throughout the manuscript and hope that everything is correct now.

Round 2

Reviewer 1 Report

The revised manuscript is acceptable for publication.